# Exploring Graph Pre-training for Aspect-based Sentiment Analysis

**Xiaoyi Bao[1], Zhongqing Wang[1]***, **Guodong Zhou[1]**
[1]Natural Language Processing Lab, Soochow University, Suzhou, China
p2213545413@outlook.com
{wangzq,gdzhou}@suda.edu.cn

## Abstract

Existing studies tend to extract the sentiment elements in a generative manner in order to avoid complex modeling. Despite their effectiveness, they ignore importance of the relationships between sentiment elements that could be crucial, making the large pre-trained generative models sub-optimal for modeling sentiment knowledge. Therefore, we introduce two pre-training paradigms to improve the generation model by exploring graph pre-training that targeting to strengthen the model in capturing the elements' relationships. Specifically, We first employ an Element-level Graph Pre-training paradigm, which is designed to improve the structure awareness of the generative model. Then, we design a Task-level Graph Pre-training paradigm to make the generative model generalizable and robust against various irregular sentiment quadruples. Extensive experiments show the superiority of our proposed method, and validate the correctness of our motivation. Our code can be found in https://github.com/HoraceXIaoyiBao/EGP4ABSA-EMNLP2023.

## 1 Introduction

Aspect-based sentiment analysis (ABSA) has drawn increasing attention in the community, which includes four fine-grained elements: aspect term, opinion term, aspect category, and opinion polarity. The first two terms exist as a raw text span in the review sentence while the remaining two are the classification result of aspect and opinion respectively. Each four mapped sentiment elements form an aspect-level sentiment quadruple. For instance, for the given review "*The apps are hard to use.*", the corresponding quadruple is (*apps, hard, Software, Negative*).

The joint extraction of quadruples is the most complex and challenging subtask among all the ABSA tasks, previous work usually formulate it as

---
* Corresponding author

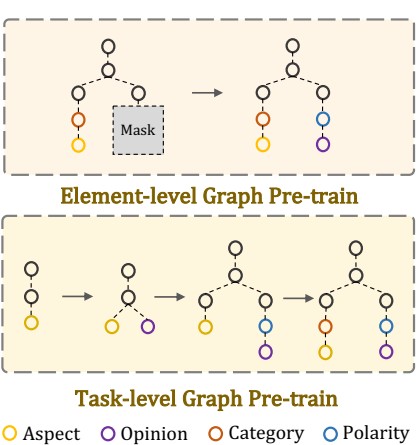

Figure 1: Two proposed pre-train paradigms.

either sequence-level (Qiu et al., 2011; Peng et al., 2020; Cai et al., 2021) or token-level classification problems (Tang et al., 2016) in joint learning or pipeline manner. However, these methods not only require sophisticated and complex modeling of sentiment elements but also suffer severely from error propagation since the overall prediction performance hinges on the accuracy of every step (Peng et al., 2020).

More recently, studies tend to tackle the ABSA problem with a unified generative approach (Zhang et al., 2021b,a; Yan et al., 2021; Bao et al., 2022). They organize the target sequence in different approaches, namely listing (Zhang et al., 2021b): "(*apps, hard, Software, Negative*)", indexing(Yan et al., 2021): "(*1,1,3,3*)", paraphrasing (Zhang et al., 2021a): "(*Software is good because apps are hard*)" or opinion tree(Bao et al., 2022): "((*Root,(Quad,( Aspect ( Software, apps ),( Opinion ( Negative, hard )))))*)". However, they ignore the importance of the relationships among elements (e.g. sentiment polarity should be identified based on opinion words, like *great* identifies a positive polarity and *disappointing* identifies a negative polarity).

In this situation, a natural question is how to

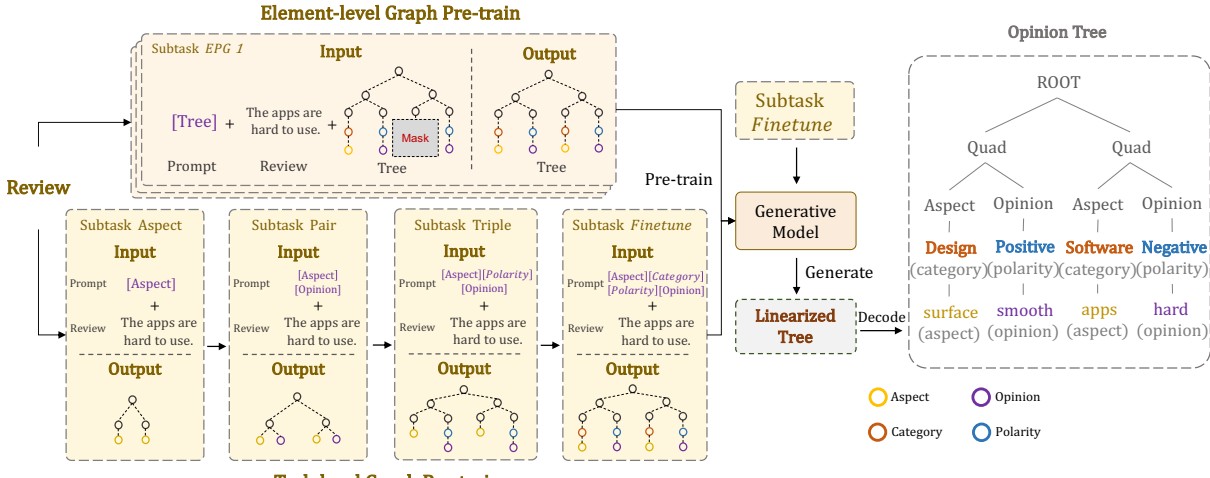

Figure 2: Overview of joint pre-training, the subtasks will be introduced in the following section. We simplify the process of Task-level Graph Pre-training for brief reading, the detailed process will be introduced in the following section.

strengthen the generative model in modeling aspect-level sentiment structure. We believe the challenges locate in two aspects. First is structural modeling: the huge gap between the pre-training and finetuning phases makes it difficult to model its succinct yet distinctive structure : certain components ( e.g. aspect term ) in sentiment structure obviously more important than others. Another challenge is the generalization and robustness of the generative model: the generative model should be generalizable and robust against irregular sentiment quadruples. It is crucial since the structure is built depending on the quadruples and the challenging scenarios in real practice are usually brought by the irregular sentiment quadruples.

In this study, we proposed two novel graph pre-training paradigms to address above challenges. As shown in Figure 1, we first introduce an optimal self-encoding method called Element-level Graph Pre-training. We abandon the traditional indiscriminate masking strategy (equally random masking every node or edge ) and depending on the characteristics of the opinion tree, adopt sentiment element level masking. Given the opinion tree of the review "*The apps are hard to use.*", only sentiment nodes (namely *apps*, *hard*, *Software*, *Negative* ) or the sub-trees they composed in the graph will be masked. In this case, this method can serve as an effective addition to structural modeling in opinion tree generation.

We then propose a Task-level Graph Pre-training paradigm, which mimics the human learning pro-cedure to learn to handle the task in stages. Specifically, we first decompose the quadruple extraction task into multiple subtasks. Each subtask corre-sponds to mapping the steps for manually building an opinion tree from scratch. Afterwards, we fea-ture a prompt-based learning strategy to separately acquire the knowledge of subtasks and finally em-ploy the learned knowledge to tackle the main task, i.e., generating the entire opinion tree. The decom-posed subtasks build fundamental knowledge of irregular sentiment quadruples for generation.

As shown in Figure 2, we then jointly pre-train the model with the two paradigms above and fine-tune the model with the $Finetune$ task. The ad-vantages of our pre-training method over previ-ous learning methods are threefold: 1) both the Element-level Graph Pre-training and Task-level Graph Pre-training are designed depending on the intrinsic characteristics of the opinion tree instead of treating it as a plain graph.2) the Element-level Graph Pre-training abandons the strategy of cap-turing the complex structure but focuses directly on the core elements. 3) the Task-level Graph Pre-training explicitly forces the model to learn the irregular quadruples with an easy-to-hard routine, making it easier for the model to learn the funda-mental knowledge required. The detailed evalua-tion shows that our model significantly advances the state-of-the-art performance on several bench-mark datasets.

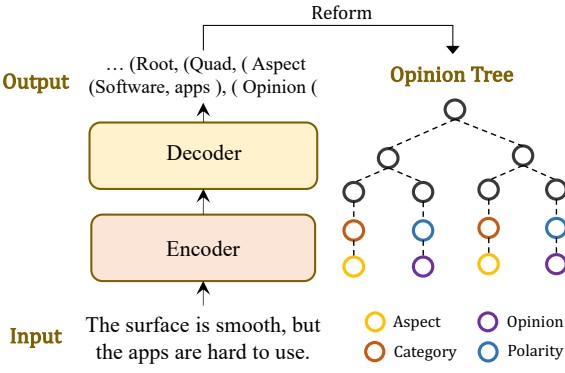

Figure 3: Opinion tree generation model.

## 2 Related Work

There are four aspect-level sentiment elements in ABSA, the various combination of these elements form the numerous sub-tasks of ABSA. The researches on ABSA generally follow a route from handling single sub-task to complex compositions of them. The starting point usually locates in the prediction of a single sentiment element, which is the target of fundamental sub-tasks, such as extracting the aspect term (Qiu et al., 2011; Tang et al., 2016; Wang et al., 2021), classifing the aspect category mentioned in the sentence (Bu et al., 2021; Hu et al., 2019), and detecting the sentiment polarity for a given aspect (Tang et al., 2016; Chen et al., 2022a; Liu et al., 2021; Seoh et al., 2021; Zhang et al., 2022).

Since the sentiment elements are naturally correlated, many studies further focus on exploring the co-extraction of sentiment elements, including aspect and opinion term extraction (Xu et al., 2020; Li et al., 2022); aspect term extraction and its polarity detection (Zhang and Qian, 2020); aspect category and polarity detection (Cai et al., 2020). Furthermore, recent studies also employed end-to-end models to extract all the sentiment elements in triplet or quadruple format (Peng et al., 2020; Wan et al., 2020; Cai et al., 2021; Zhang et al., 2021a; Chen et al., 2022b; Mukherjee et al., 2021).

More recently, studies tend to design a unified framework to extract quadruples at one stop with pre-trained encoder-decoder language models, achieving great improvements in ABSA (Zhang et al., 2021a). The target sequence of them is formed by either class index (Yan et al., 2021) or the desired sentiment element (Zhang et al., 2021b). OTG (Bao et al., 2022) addressed the importance of semantic correlations among sentiment elements, proposed a sentiment tree structure called opinion tree, and employed generative model to extract the linearized tree. However, the generative model is pre-trained to solve textual sequence tasks(e.g. masked language model) but finetuned for structure generation, between which exists a huge gap, making generative models sub-optimal for modeling structural knowledge.

Different from previous studies, we introduce two pre-training paradigms for opinion tree generation without treating it as a plain graph. To our knowledge, we are the first to consider designing methods depending on the intrinsic characteristics of the opinion tree.

## 3 Opinion Tree Generation Model

In this section, we introduce the basic opinion tree generation model we employed to generate in the pre-train and finetune phases, along with the objective functions and training.

### 3.1 Opinion Tree Construction

For further strengthen the relationship between elements, we build a structure called opinion tree, which aims to jointly model all sentiment elements in a tree for a given review sentence. The opinion tree can be considered as a semantic representation in order to better represent the structure of sentiment elements. Inside the opinion tree, each sentiment element would be connected with another node as either the child or parent relation to represent the crucial relationship.

As shown in Figure 3, we construct the opinion tree using a rooted directed acyclic graph, including nodes of aspect, opinion, category, and polarity, along with the semantic relations between them. After that, we linearize the opinion tree to the target sequence via depth-first traversal.

### 3.2 Generation Model

We employ the pre-trained language model T5 (Raffel et al., 2020) to generate the linearized opinion tree. As shown in Figure 3, it is an encoder-decoder architecture model, the input would be the raw review and the output is linearized opinion tree. Given the token sequence $x = x_1, ..., x_{|x|}$ as input, the sequence-to-sequence model outputs the linearized representation $y = y_1, ..., y_{|y|}$. To this end, the sequence-to-sequence model first computes the hidden vector representation:

$$H = (x_1, ..., x_{|x|}) \tag{1}$$

| Subtask | Input | | | Subtask |
| --- | --- | --- | --- | --- |
| | Prompt | Review | Tree | |
| EGP1 | [Tree] | The apps are hard to use. | (Root,(Quad, ...<Mask>,...) | (Root,(Quad, ...(Negative,hard) |
| EGP2 | [Sentence] | The apps <Mask> use. | <Mask> | The apps are hard to use. |
| EGP3 | [Sentence] | The apps <Mask> use. | (Root,(Quad, ...(Negative, hard) | The apps are hard to use. |
| EGP4 | [Tree] | The apps <Mask> use. | (Root,(Quad, ...<Mask>,...) | (Root,(Quad, ...(Negative,hard) |
| EGP5 | [Sentence] | The apps <Mask> use. | (Root,(Quad, ...<Mask>,...) | The apps are hard to use. |

Table 1: Subtasks of Element-level Graph Pre-training.

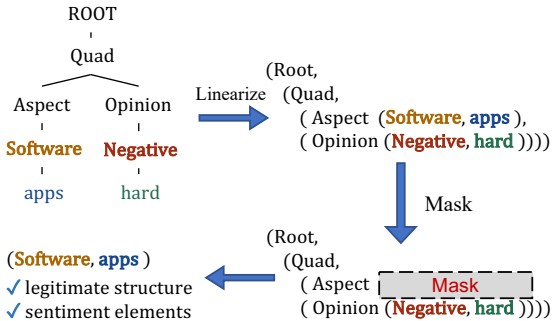

Figure 4: Example of element-level graph masking.

After the input token sequence is encoded, the decoder predicts the output sequence token-by-token with the sequential input tokens' hidden vectors. At the $i$-th step of generation, the self-attention decoder predicts the $i$-th token $y_i$ in the linearized form, and decoder state $h_i^d$ as:

$$y_i, h_i^d = ([H; h_1^d, ..., h_{i-1}^d], y_{i-1}) \quad (2)$$

The conditional probability of the whole output sequence $p(y|x)$ is progressively combined by the probability of each step $p(y_i|y_{<i}, x)$:

$$p(y|x) = \prod_{i=1}^{|y|} p(y_i|y_{<i}, x) \quad (3)$$

where $y_{<i} = y_1...y_{i-1}$, and $p(y_i|y_{<i}, x)$ are the probabilities over target vocabulary $V$.

The objective functions is to maximize the output linearized opinion tree $X_T$ probability given the review sentence $X_O$. Therefore, we optimize the negative log-likelihood loss function:

$$\mathcal{L} = -\frac{1}{|\tau|} \sum_{(X_O, X_T) \in \tau} \log p(X_T|X_O; \theta) \quad (4)$$

where $\theta$ is the model parameters, and $(X_O, X_T)$ is a (*sentence, tree*) pair in training set $\tau$, then

$$\log p(X_T|X_O; \theta) =$$
$$= \sum_{i=1}^{n} \log p(x_T^i|x_T^1, x_T^2, ...x_T^{i-1}, X_O; \theta) \quad (5)$$

where $p(x_T^i|x_T^1, x_T^2, ...x_T^{i-1}, X_O; \theta)$ is calculated by the decoder.

## 4 Pre-training Paradigms

In this study, we introduce two pre-training paradigms for opinion tree generation. As shown in Figure 2, the two paradigms and finetune task share the same input format with a joint input of prompt, encoded text and tree, each method consists of a set of subtasks focus on respective training targets. The combination of subtasks forms the joint pre-training in our work, we will introduce the paradigms first in this section.

### 4.1 Element-level Graph Pre-training

The opinion tree is directly composed of subtrees that represent respective quadruples, this naturally decides the noteworthy information must locate within the aspect-level sentiment element instead of the other parts of the opinion tree, which could be other structure nodes. For instance, for a linearized opinion tree "*(Root,(Quad,(Aspect (Software, apps),(Opinion (Negative, hard)*", the indiscriminate masking may mask a sub-sequence "*(Opinion (*" that: 1) logically can not be reform into a valid structure due to the non-closing brackets. 2) contains nodes (e.g."*Opinion*" ) not included in the crucial sentiment elements.

On the other hand, our Element-level Graph Pre-training paradigm masks aspect-level element nodes (including aspect term, opinion term, aspect category, and opinion polarity) in the opinion tree, as shown in Figure 4, the masked sequence "*(Software, apps )*" represent legitimate struct and covers core sentiment element only. If continuous nodes are masked, the corresponding sub-graph will be masked as a whole. The method can not only make sure the masked node are crucial sentiment elements but also guarantee the corresponding sub-sequence is logically legitimate.

With the element-level graph mask strategy introduced above, we propose a set of pre-training

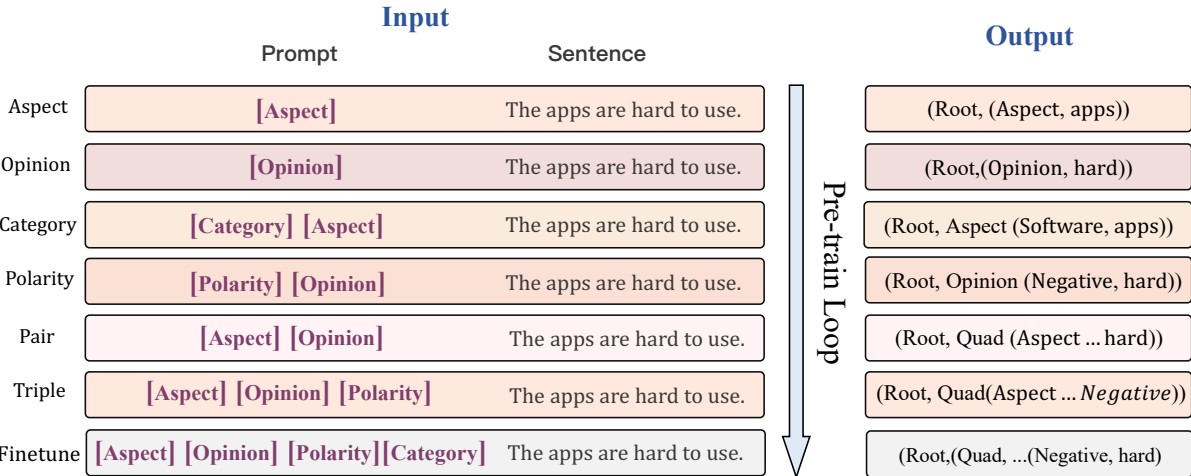

Figure 5: Subtasks of Task-level Graph Pre-training paradigm. Note the finetune task has been added into the pre-training phase.

| Epoch range | Tree mask rate | Text mask rate |
|---|---|---|
| $0\% \sim 25\%$ | 0.25 | 0.15 |
| $25\% \sim 50\%$ | 0.3 | 0.15 |
| $50\% \sim 75\%$ | 0.35 | 0.15 |
| $75\% \sim 100\%$ | 0.4 | 0.15 |

Table 2: Dynamic masking rate.

subtasks. The inputs would be a concat of a prompt, a sentence, and an opinion tree. The sentence and tree will be masked with different masking rates while the prompt illustrates the output target, either the sentence or tree. For a given review $s = (x_1, x_2, ...x_{n-1}, x_n)$ and linearized tree $t = (t_1, t_2, ...t_{n-1}, t_n)$, We design the 5 subtasks in the Element-level Graph Pre-training paradigm, which can be found in Table 1. Among which, $EPG1$ and $EPG4$ are designed to help the model generate the complete tree $t$ by adding text information while $EPG2$, $EPG3$ and $EPG5$ help the model to generate the full review $s$ by adding the structural information.

To further emphasize the interaction between the pre-training and finetune phases, we designed a dynamic masking rate for Element-level Graph Pre-training paradigms: a small masking rate is used in the initial phase, and then the masking rate increases with training rounds, so that at the end of pre-training, all partially masked pre-training tasks be very close to the finetune tasks (which can be considered as 100% masking rate), the specific masking rate is shown in Table 2. Note our masking rate obviously lower than previous work (Bai et al., 2022), that is because recovering a nearly all-masked text from an opinion tree is unreasonable since opinion tree contains limited information as we discussed before.

## 4.2 Task-level Graph Pre-training

Inspired by the human-learning process we propose a Task-level Graph Pre-training paradigm, whose subtasks follow the routine of human learning procedure to learn to build the opinion tree from scratch. Specifically, we first decompose the quadruple extraction task into multiple subtasks. Each subtask corresponds to mapping the steps for manually building an opinion tree from scratch. The paradigm consists of six subtasks, four ($Aspect$, $Opinion$, $Category$, $Polarity$) of which extract sentiment structure as the fundamental knowledge for building an opinion tree, the rest ($Pair$, $Triple$) target the intermediate state of the procedure with co-extraction. The subtasks and the corresponding steps of building can be found in Appendix A. In this case, we force the model to focus directly on irregular cases with a gradual process to build fundamental knowledge for OTG. The inputs of Task-level Graph Pre-training are similar to the previous paradigm, which would be a concat of a prompt and a sentence. Then the subtasks in Task-level Graph Pre-training paradigm can be given as shown in Figure 5.

## 4.3 Joint Pre-training

We use a joint pre-training method to combine the advantages of the Element-level Graph Pre-training paradigm and Task-level Graph Pre-training paradigms. In addition, we include the

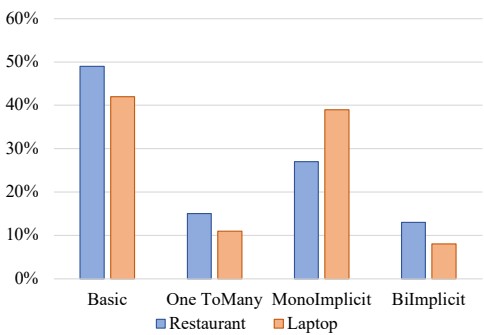

Figure 6: Statistic of regular and irregular situations of opinion trees.

finetune task $Finetune$ in the pre-train phase for narrowing the gap between two phases and avoiding overfitting. During pre-training, the model will be cyclically trained in the order of a loop started with the subtasks of the Element-level Graph Pre-training, followed by Task-level Graph Pre-training, the gradient will be updated after accumulating the loss in each epoch. After that, we save the model weights and finetune the model with finetune task $Finetune$.

## 5 Experiments

In this section, we introduce the datasets used for evaluation and the baseline methods employed for comparison. We then report the experimental results conducted from different perspectives, and analyze the effectiveness of the proposed model with different factors.

### 5.1 Setting

In this study, we use ACOS dataset (Cai et al., 2021) for our experiments. Following the setting from (Cai et al., 2021), we divide the original dataset into a training set, a validation set, and a testing set. In addition, we choose 20,000 sentences from Yelp[1], and 20,000 sentences from the laptop domain in Amazon[2] to pre-train the opinion tree generation model, the sentences are annotated by the OTG model without pre-training.

Following the setting of Bao et al. (2023), we divide the quadruples into 4 types, apart from the basic situation, there are 3 irregular situations: **One-to-Many**, **Mono-Implicit** and **Bi-Implicit**. The statistic can be found in Figure 6.

---

[1]https://www.yelp.com/dataset
[2]http://jmcauley.ucsd.edu/data/amazon/

We employ T5[3] and fine-tune its parameters for our opinion tree generation model. We tune the parameters of our models by grid searching on the validation dataset. We select the best models by early stopping using the Accuracy results on the validation dataset. The model parameters are optimized by Adam (Kingma and Ba, 2015), the learning rate of pre-training and finetuning is 3e-5 and 1e-4 respectively. The batch size is 16. Our experiments are carried out with an Nvidia RTX 3090 GPU. The experimental results are obtained by averaging ten runs with random initialization.

In evaluation, a quadruple is viewed as correct if and only if the four elements, as well as their combination, are exactly the same as those in the gold quadruple. On this basis, we calculate the Precision and Recall, and use F1 score as the final evaluation metric for aspect sentiment quadruple extraction (Cai et al., 2021; Zhang et al., 2021a).

### 5.2 Main Results

We compare the proposed method with several classification-based aspect-based sentiment analysis models, including, *DP* (Qiu et al., 2011), *JET* (Xu et al., 2020), *TAS-BERT* (Wan et al., 2020) and *Extract-Classify* (Cai et al., 2021). In addition, generative models are also compared, such as *BARTABSA* (Yan et al., 2021), *GAS* (Zhang et al., 2021b), *Paraphrase* (Zhang et al., 2021a),*TODA* (Hu et al., 2022), *Seq2Path* (Mao et al., 2022) and *OTG* (Bao et al., 2022).[4].

Particularly, we build two Large Language Model (LLM) baselines: *ChatGPT*[5] is a sibling model to InstructGPT (Ouyang et al., 2022), which is trained to follow instruction in a prompt and provide a detailed response. We ask it to generate all the sentiment elements from the input review sentences. *LLaMA*[6] (Touvron et al., 2023) is a collection of foundation language models, these models are trained on trillions of tokens, and have shown that it is possible to train state-of-the-art models using publicly available datasets exclusively. We use LLaMA-7B, and fine-tune it on the ABSA dataset.

As shown in Table 3, we find that generative models outperform previous classification-based methods and the structural generative method sur-

---

[3]T5$_{base}$, https://huggingface.co/transformers/model_doc/t5.html
[4]We directly adopt the result from Bao et al. (2022)
[5]https://openai.com/blog/chatgpt.
[6]https://huggingface.co/docs/transformers/main/model_doc/llama.

| Method | Restaurant | | | Laptop | | |
|---|---|---|---|---|---|---|
| | P. | R. | F1. | P. | R. | F1. |
| DP | 0.3467 | 0.1508 | 0.2104 | 0.1304 | 0.0057 | 0.0800 |
| JET | 0.5981 | 0.2894 | 0.3901 | 0.4452 | 0.1625 | 0.2381 |
| TAS-BERT | 0.2629 | 0.4629 | 0.3353 | **0.4715** | 0.1922 | 0.2731 |
| Extract-Classify | 0.3854 | 0.5296 | 0.4461 | 0.4556 | 0.2948 | 0.3580 |
| BARTABSA | 0.5662 | 0.5535 | 0.5598 | 0.4165 | 0.4046 | 0.4105 |
| GAS | 0.6069 | 0.5852 | 0.5959 | 0.4160 | 0.4275 | 0.4217 |
| Paraphrase | 0.5898 | 0.5911 | 0.5904 | 0.4177 | 0.4504 | 0.4334 |
| TODA | 0.5904 | 0.6029 | 0.5966 | 0.4359 | 0.4367 | 0.4363 |
| Seq2Path | 0.6029 | 0.5961 | 0.5995 | 0.4448 | 0.4375 | 0.4411 |
| ChatGPT | 0.5014 | 0.3625 | 0.4207 | 0.4492 | 0.3123 | 0.3541 |
| LLaMA | 0.5963 | 0.6097 | 0.6029 | 0.4461 | 0.4392 | 0.4426 |
| OTG | 0.6138 | 0.6190 | 0.6164 | 0.4408 | 0.4381 | 0.4394 |
| Ours | **0.6486** | **0.6297** | **0.6390** | 0.4523 | **0.4523** | **0.4512** |

Table 3: Comparison with baselines.

passes non-structural methods, this indicates that semantic structure does contribute to quadruple extraction. Meanwhile, our proposed model outperforms all the previous studies significantly ($p <$ 0.05), which has an advantage of 2.36% and 0.92% in Restaurant and Laptop domain respectively. The result shows that the proposed joint pre-training is effective in modeling tree structural constraints for generative model, while the large gap between pre-training and finetuning significantly encumbers previous systems. Furthermore, the results also indicate the effectiveness of our Element-level Graph Pre-training and Task Decomposition paradigms, which are used to unify the pre-train and finetune task with special task designs depending on the intrinsic characteristics of the opinion tree instead of treating it as a plain graph.

## 6 Analysis and Discussion

In this section, we first give some analysis and discussion to show the influence of Element-level Graph Pre-training (EGP) and Task-level Graph Pre-training (TGP) paradigms. After that, we will investigate our search over masking rate, the influence of pre-training subtasks.

### 6.1 Influence of Different Factors

We first investigate the difference between the two paradigms, from Table 4 we can find, all the paradigms are beneficial to extract the opinion tree. Among which TGP paradigm's contribution outperforms EGP paradigm, the removal of TGP cause an

| Method | Restaurant | Laptop |
|---|---|---|
| Ours | 0.6390 | 0.4512 |
| - EGP | 0.6339 | 0.4490 |
| - TGP | 0.6334 | 0.4463 |
| - EGP& TGP | 0.6164 | 0.4393 |

Table 4: Impact of pre-training paradigm.

| Method | Restaurant | Laptop |
|---|---|---|
| OTG | 0.6164 | 0.4394 |
| + Indiscriminate | 0.6287 | 0.4423 |
| + $Finetune$ | 0.6211 | 0.4411 |
| + $EGP1, Finetune$ | 0.6243 | 0.4413 |
| + $EGP1, Finetune$ $EGP2, EGP3$ | 0.6276 | 0.4421 |
| + All $EGPs$ | 0.6334 | 0.4463 |
| Ours | 0.6390 | 0.4512 |

Table 5: Impact of subtasks in Element-level Graph Pre-training paradigm.

avgerage drop of 0.52% while EGP's cause 0.21%, this may due to the generalization and robustness being more effective than the structural association.

### 6.2 Effect of Element-level Graph Pre-training

Under the setting of our element-level masking design for graph pre-train, previous graph-masking strategies can be classified into the indiscriminate paradigm, which means indiscriminately masking random nodes and words in tree or text. In

| Method | OTG | Ours | |
|---|---|---|---|
| Basic | 0.6517 | 0.6610 | + 0.93% |
| OneToMany | 0.4503 | 0.4720 | + 2.17% |
| MonoImplicit | 0.4035 | 0.4261 | + 2.26% |
| BiImplicit | 0.4184 | 0.4341 | + 1.57% |

Table 6: The average performance of different situations in Restaurant and Laptop domain.

| Method | Restaurant | Laptop |
|---|---|---|
| OTG | 0.6164 | 0.4394 |
| + $Aspect$ | 0.6251 | 0.4439 |
| + $Aspect, Opinion$ | 0.6275 | 0.4447 |
| + $Aspect, Opinion,$ $Pair, Triple$ | 0.6294 | 0.4473 |
| + $Aspect, Opinion, category$ $polarity, Pair, Triple$ | 0.6294 | 0.4473 |
| + $Aspect, Opinion, category$ $polarity, Pair, Triple$ $finetune$ | 0.6339 | 0.4490 |

Table 7: Impact of subtasks in Task-level Graph Pre-training paradigm.

this situation, there will be one intuitive question: *Whether the element-level masking design does achieve a performance better than the indiscriminate paradigm as we expect?*

We investigate this question by employing ablation experiments. We first design an indiscriminate paradigm under similar settings, then we give the performance of using different paradigms in Table 5. As we can see, our element-level paradigm outperforms the indiscriminate paradigm, this result shows the superiority of our element-level masking design, and also validated our motivation: for target graphs that contain limited knowledge like opinion tree, indiscriminate masking strategies would be sub-optimal and fine-grained masking should be adopted.

We then investigate the impact of subtasks in EGP paradigm. We add the subtasks in paradigm gradually. As we can see in Table 5, the subtask pair of $EPG5$ and $EPG4$ (+All $EGPs$) contributes the most to the performance (0.58% and 0.42% in each domain respectively), which aims to integrate the complementary information from both formations to generate text and tree respectively, indicating the significance of the complementary association.

### 6.3 Effect of Task-level Graph Pre-training

As shown in Table 6, the OTG model obviously be short in its generalization and robustness against irregular sentiment quadruples when compared with the basic situation. Thus we mimic the human learning procedure for building an opinion tree from scratch with Task-level Graph Pre-training to strengthen its fundamental knowledge.

We investigate the paradigm's effect by comparing the model's performance on each irregular quadruple situation. As shown in Table 6 , our model's improvement in all of the irregular classes surpasses the basic situation when compared with OTG. This result indicates that our pre-train method significantly improves the model's performance with a burst in generalization and robustness against irregular sentiment quadruples, which accomplish the foundation for building an opinion tree and should be taken into consideration apart from improving the structural awareness.

We then investigate the impact of subtasks in TGP paradigm. We remove the subtasks in the paradigms gradually. Table 7 shows the result for Task Decomposition paradigm: the contributions of subtasks stay in a similar scope, among which the $Aspect$ surpasses others with a tiny gap, this may due to the lower implicit rate of aspect terms[7].

In addition, all the subtasks are beneficial to extract the opinion tree. It is worth noting that, the participation of finetune task $Finetune$ demonstrates an obviously positive effect in both paradigms, which improves two domains with an average of 0.31%, this phenomenon gives us a conclusion that adding the finetune task in the pre-train phase is an effective solution for narrowing the gap between them.

## 7 Conclusion

In this study, we propose two novel pre-train paradigms for opinion tree generation, which are designed depending on the intrinsic characteristics of the opinion tree. Specifically, the Element-level Graph Pre-training paradigm abandons the strategy of capturing the complex structure but focuses directly on the core elements. While the Task-level Graph Pre-training explicitly focuses on improving the generalization and robustness against irregular quadruples with an easy-to-hard routine. Furthermore, we explore a dynamic masking rate and a cyclical train method for jointly combining the pre-training paradigms in order to bridge the gap between the pre-training and finetuning phases in modeling structural knowledge.

---

[7]The average implicit rate of aspect term and opinion term is 22.63% and 24.19% respectively

Experimental results show that our proposed model can achieve state-of-the-art performance in ABSA. In addition, the results also validate that, for target graphs that contain certain knowledge like opinion tree, the improving strategy should be made based on the intrinsic characteristics of the structure instead of treating it as a plain graph.

## Limitations

The limitations of our work can be stated from three perspectives. First, our pre-training method contains many subtasks that will consume vast computational cost during pre-train (the inference cost will not change). If possible, further work should try to explore a time-saving pre-training method. Secondly, more tasks could be further explored, including cross-domain and cross-lingo sentiment analysis tasks. Finally, we focus on opinion tree generation in one major language. The performance of other languages remains unknown.

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

## A  Building Procedure

In Task-level Graph Pre-training paradigm, the subtasks are set to follow the routine of building the opinion tree from scratch. For building an opinion tree manually, humans often learn to find fundamental elements, such as aspects or opinions, followed by finding the corresponding classification result such as category and polarity to build a single quadruple unit, then composing multiple units to fulfill a more challenging goal, i.e., writing the entire opinion tree.

Based on the process introduced, we design the subtasks in Task-level Graph Pre-training paradigm. Each subtask corresponds to mapping the steps for manually building an opinion tree from scratch. The paradigm consists of six subtasks: *Aspect*, *Opinion*, *Category*, *Polarity*, *Pair* and *Triple*. Their prompts and target graph can be found in Figure 7. Among which, *Aspect* and *Opinion* focus on searching the basic elements of each quadruple:

- *Aspect*: Extract all the aspect terms in the review in the form of a tree, Figure 7 (a).

- *Opinion*: Extract all the Opinion terms in the review in the form of a tree, Figure 7 (b).

*Category* and *Polarity* further explore the classification results with the corresponding basic elements:

- *Category*: On the base of *Aspect*, extract the category classification result of the aspect terms in the review in the form of a tree, Figure 7 (c).

- *Polarity*: On the base of *Opinion*, extract the polarity classification result of the opinion terms in the review in the form of a tree, Figure 7 (d).

*Pair* and *Triple* fulfill the mapping between quadruples.

- *Pair*: On the base of *Aspect* and *Opinion*, map the corresponding aspect term and opinion term within a quadruple, Figure 7 (e).

- *Triple*: On the base of *Aspect* and *Polarity*, map the corresponding aspect term and opinion term and polarity within a quadruple, Figure 7 (f).

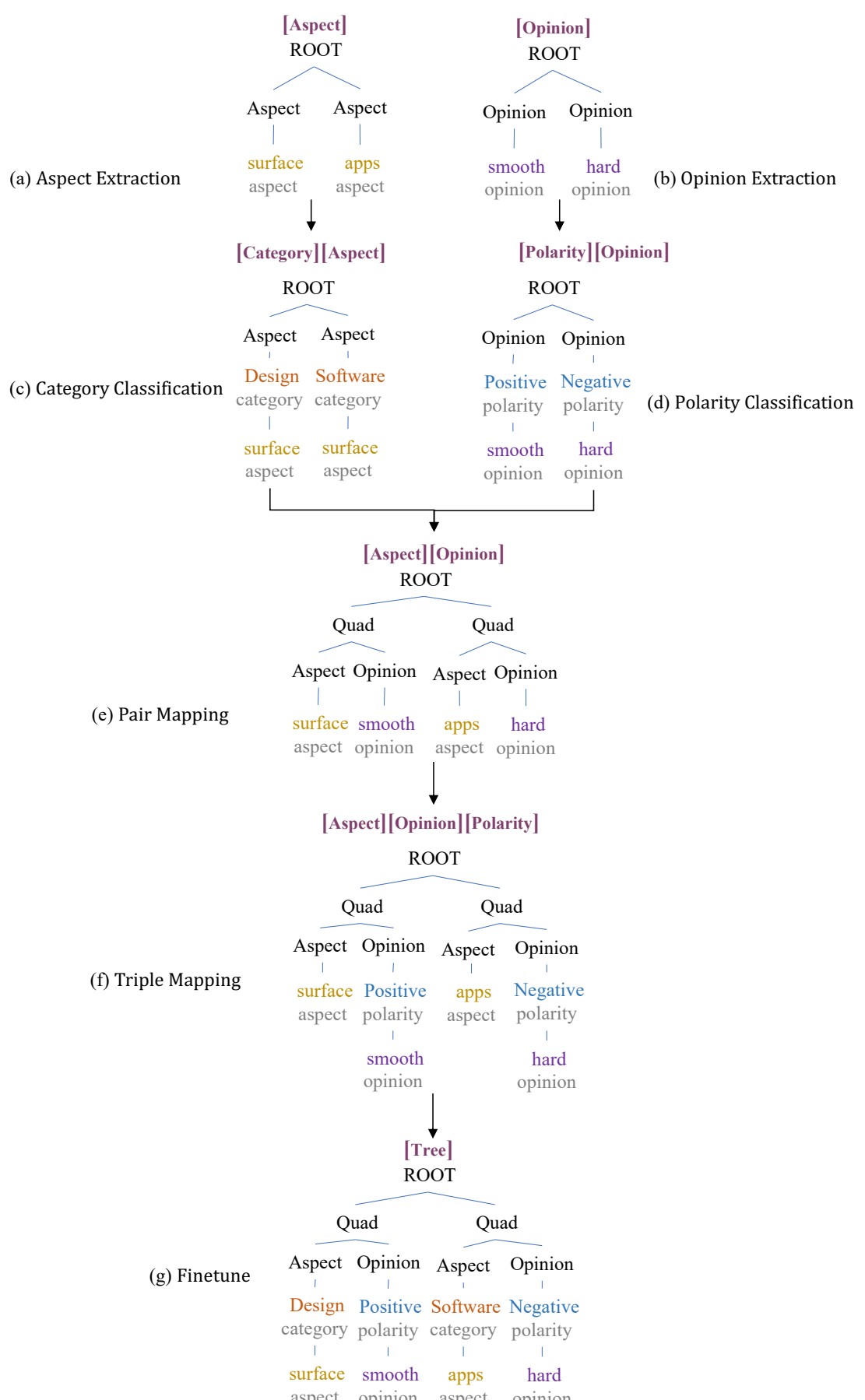

Figure 7: Building procedure of subtasks in Task-level Graph Pre-training.