# OpenReview forum: "Exploring Graph Pre-training for Aspect-based Sentiment Analysis"
_EMNLP/2023/Conference — EMNLP 2023 Findings_

### Official Review · Reviewer_LibN · 2023-08-02

**Soundness:** 4

**Excitement:**

4: Strong: This paper deepens the understanding of some phenomenon or lowers the barriers to an existing research direction.

**Paper Topic And Main Contributions:**

This paper introduce two pre-training paradigms to improve the generation model for Aspect-based sentiment analysis.

1. This paper employ an Element-level Graph Pretraining paradigm, which is designed to improve the structure awareness of the generative model.
2. This paper design a Task-level Graph Pre-training paradigm to make the generative model generalizable and robust against various irregular sentiment quadruples.

The experiment results show the superiority of our proposed method, validate the correctness of our motivation.

**Reasons To Accept:**

This paper proposes a reasonable pre-training method to improve the model, and the experimental results verify the effectiveness of the method. The implementation described by the author is rich in details and reproducible.

**Reasons To Reject:**

This paper only use ACOS dataset (Cai et al., 2021) for experiments. Experiments on more datasets will make this paper more valuable.

**Reproducibility:**

5: Could easily reproduce the results.

**Reviewer Confidence:**

4: Quite sure. I tried to check the important points carefully. It's unlikely, though conceivable, that I missed something that should affect my ratings.

---

> ### Author Rebuttal · Authors · 2023-08-27
>
> #### Dear reviewer,
>
> #### We appreciate your valuable reviews, we will respond to your review one by one as follows:
>
> > This paper only use ACOS dataset (Cai et al., 2021) for experiments. Experiments on more datasets will make this paper more valuable.
>
> #### **Response:** For the aspect-level sentiment task, the ACOS dataset is the only dataset that contains the full elements(aspect, opinion, category, polarity) we are motivated to focus on with a refined element-level pretraining method. Besides, the ACOS dataset has already covered the previous SemEval Res/Lap 14/15/16 dataset, making it comprehensive for validating the method's performance.

---

### Official Review · Reviewer_JuKD · 2023-08-04

**Typos Grammar Style And Presentation Improvements:** 1. line 021, "validate" should be "an…
**Soundness:** 3

**Excitement:**

3: Ambivalent: It has merits (e.g., it reports state-of-the-art results, the idea is nice), but there are key weaknesses (e.g., it describes incremental work), and it can significantly benefit from another round of revision. However, I won't object to accepting it if my co-reviewers champion it.

**Paper Topic And Main Contributions:**

This paper focuses on the ABSA task in a generational way. The authors present two pre-training paradigms for opinion tree generation, including Element-level and Task-level Graph Pre-training. Experimental results validate the effectiveness of the proposed method.

**Questions For The Authors:**

1. In Line 125, the paper claimed "several benchmark datasets", maybe the "ACOS dataset" is better.
2. In lines 517-520, the authors claimed, "the Task-level Graph Pre-training explicitly focuses on improving the generalization and robustness ..." . Regarding experiments, generalization, and robustness have not been presented yet. Perhaps experiments with more tasks or more datasets could be added.
3. In Fig. 5, for the prompt of different sub-task, is it still effective to use the numbers 0-5 as prompts for different subtasks? As the displayed prompt is just a tag.


**Reasons To Accept:**

1. The authors present two graph-based pre-training paradigms in detail;
2. This paper gives a new perspective on the ABSA task in a pre-trained generational manner.


**Reasons To Reject:**

1. Experiments were performed on the ACOS dataset only, and more datasets can be added.
2. As claimed in the Limitation, the computational cost of so many tasks during pre-training is huge.

**Reproducibility:**

3: Could reproduce the results with some difficulty. The settings of parameters are underspecified or subjectively determined; the training/evaluation data are not widely available.

**Reviewer Confidence:**

5: Positive that my evaluation is correct. I read the paper very carefully and I am very familiar with related work.

---

> ### Author Rebuttal · Authors · 2023-08-27
>
> #### Dear reviewer,
>
> #### We appreciate your valuable reviews, we will respond to your review one by one as follows:
>
> > 1. Experiments were performed on the ACOS dataset only, and more datasets can be added.
>
> #### **Response:** For the aspect-level sentiment task, the ACOS dataset is the only dataset that contains the full elements(aspect, opinion, category, polarity) we are motivated to focus on with a refined element-level pretraining method. Besides, the ACOS dataset has already covered the previous SemEval Res/Lap 14/15/16 dataset, making it comprehensive for validating the method's performance.
>
> > 2. As claimed in the Limitation, the computational cost of so many tasks during pre-training is huge.
>
> #### **Response:** Compared with other pretraining methods such as the joint pre-training method used in [Bao IJCAI 2022], which involved not only ABSA annotations but also AMR and dependence trees, our method has already saved the computational cost as it only needs the in-task annotation cost.
>
> - [Bao IJCAI 2022] Aspect-based Sentiment Analysis with Opinion Tree Generation, IJCAI 2022 (OTG method)
>
> > In Line 125, the paper claimed "several benchmark datasets", maybe the "ACOS dataset" is better.
> #### **Response:** We appreciate your suggestion, and we will correct it once accepted.
>
>
> > In lines 517-520, the authors claimed, "the Task-level Graph Pre-training explicitly focuses on improving the generalization and robustness ..." . Regarding experiments, generalization, and robustness have not been presented yet. Perhaps experiments with more tasks or more datasets could be added.
> #### **Response:**
> - The improvement of the generalization and robustness against irregular quadruples has been shown in Table 6 and explained in Section 6.3.
>
> - The Task-level Graph Pre-training can explicitly improve the model's generalization and robustness when facing the irregular quadruples as it builds the opinion tree following the build-from-scratch procedure, which means every irregular composition has been emphasized during the pre-training, such as one-to-many will be focused on in the subtask shown in Fig 5 line 5.
>
>
> >In Fig. 5, for the prompt of different sub-task, is it still effective to use the numbers 0-5 as prompts for different subtasks? As the displayed prompt is just a tag.
> #### **Response:** We haven't done the experiment yet, we believe even if it works, it will still cause a slight decrease in performance since the pre-trained language model will be unable to catch the semantic relationship with the prompts and specific tasks.

---

### Official Review · Reviewer_Jvv2 · 2023-08-05

**Soundness:** 3

**Excitement:**

3: Ambivalent: It has merits (e.g., it reports state-of-the-art results, the idea is nice), but there are key weaknesses (e.g., it describes incremental work), and it can significantly benefit from another round of revision. However, I won't object to accepting it if my co-reviewers champion it.

**Paper Topic And Main Contributions:**

This paper proposes two pre-training paradigms on the opinion tree to improve the generation model for aspect-based sentiment analysis (ABSA) tasks, i.e., element-level graph pre-training and task-level graph pre-training. The Element-level Graph Pre-training focuses directly on the core sentiment elements of the ABSA tasks, and it uses a sentiment element-level masking strategy. The Task-level Graph Pre-training decomposes the quadruple extraction task into multiple subtasks for building an opinion tree from scratch and employs the learned knowledge of each subtask for the main task.

**Questions For The Authors:**

1. In the main results, LLaMa-7B prominently performs better than ChatGPT. Generally, ChatGPT prominently outperforms LLaMa-7B, and 7B size is also not large enough. The main results show that after fine-tuning, LLaMa-7B can prominently performs better than ChatGPT. Could you introduce more details about the implementations and discuss more? e.g., the fine-tuning settings, the prompts used to these LLMs, should we use fine-tuned smaller LLMs rather than larger LLMs without fine-tuning?, etc.

**Reasons To Accept:**

1. The main idea is reasonable, and the manuscript is well-organized and presented.

2. The Element-level Graph Pre-training paradigm focuses directly on the core elements (aspect terms, opinion words, etc.) enabling the capture of the relationships between core sentiment elements.

3. The Task-level Graph Pre-training paradigm decomposes the quadruple extraction task into multiple subtasks to help the main task.


**Reasons To Reject:**

1. An incremental study of this topic based on previous works.
- [Bao IJCAI 2022] Aspect-based Sentiment Analysis with Opinion Tree Generation, IJCAI 2022 (OTG method)
- [Bao ACL Finding 2023] Opinion Tree Parsing for Aspect-based Sentiment Analysis, Findings of ACL 2023

The techniques involved in the proposed framework have been commonly used for ABSA, including graph pre-training, opinion tree generation and so on, and it seems not surprising enough to combine them together. The experimental results only show the performance can be improved, but lack of the explanations of why the performance can be improved.


2. The experimental results are not exactly convincing, by comparing with the main results in [Bao IJCAI 2022] and [Bao ACL Finding 2023]. For example,

- For the scores of OTG method, [Bao ACL Finding 2023] < this paper < [Bao IJCAI 2022]. Note that this is a significant difference, for example, on the Restaurant dataset, for F1 score, [Bao ACL Finding 2023] 0.6040 < this paper 0.6164 < [Bao IJCAI 2022] 0.6283; on the laptop dataset, for F1 score, [Bao ACL Finding 2023] 0.3998 < this paper 0.4394 < [Bao IJCAI 2022] 0.4544

- On the laptop dataset, for F1 score, although the scores of OTG method in this paper 0.4394 < the scores of proposed method in this paper 0.4512; The scores of OTG method in [Bao IJCAI 2022] 0.4544 > the scores of proposed method in this paper 0.4512;

There are also other significant differences on the performance of the baseline methods in these papers.


3. It could be convincing to discuss case studies and error studies to highlight the effectiveness of each proposed component. For example, this paper mentions that the Element-level Graph Pre-training abandons the strategy of capturing the complex structure but focuses directly on the core elements. However, without case study, it is less convincing to figure it out. An example of case study can be found in “Graph pre-training for AMR parsing and generation”.



**Reproducibility:**

3: Could reproduce the results with some difficulty. The settings of parameters are underspecified or subjectively determined; the training/evaluation data are not widely available.

**Reviewer Confidence:**

3: Pretty sure, but there's a chance I missed something. Although I have a good feel for this area in general, I did not carefully check the paper's details, e.g., the math, experimental design, or novelty.

**Typos Grammar Style And Presentation Improvements:**

- In L320, there is no explanation for the acronym "OTG".

- In Section 6.1, a typo in the sentence "Among which TGP paradigm’s contribution outperforms TGP paradigm,"

---

> ### Author Rebuttal · Authors · 2023-08-27
>
> #### Dear reviewer,
>
> #### We really appreciate your valuable reviews, we will respond to your review one by one as follows:
>
> > 1. The techniques involved in the proposed framework have been commonly used for ABSA, including graph pre-training, opinion tree generation and so on, and it seems not surprising enough to combine them together. The experimental results only show the performance can be improved, but lack of the explanations of why the performance can be improved.
>
> #### **Response:** The motivation of our work is applying the pretraining method to different levels of ABSA graph while other graph pretraining works stick to treating the structured sequence as plain text, which is an obviously rough method that ignores the intrinsic characteristic of the graph itself. Hence, a more refined strategy regarding the target graph is the work we are motivated to do and contributes to the improvement of the performance.
>
> > 2. The experimental results are not exactly convincing, by comparing with the main results in [Bao IJCAI 2022] and [Bao ACL Finding 2023]. For example,
> > - For the scores of OTG method, [Bao ACL Finding 2023] < this paper < [Bao IJCAI 2022]. Note that this is a significant difference, for example, on the Restaurant dataset, for F1 score, [Bao ACL Finding 2023] 0.6040 < this paper 0.6164 < [Bao IJCAI 2022] 0.6283; on the laptop dataset, for F1 score, [Bao ACL Finding 2023] 0.3998 < this paper 0.4394 < [Bao IJCAI 2022] 0.4544
> > - On the laptop dataset, for F1 score, although the scores of OTG method in this paper 0.4394 < the scores of proposed method in this paper 0.4512; The scores of OTG method in [Bao IJCAI 2022] 0.4544 > the scores of proposed method in this paper 0.4512;
>
> #### **Response:**
>
> - The difference between the OTG in [Bao IJCAI 2022] and  [Bao ACL Finding 2023] is because of their datasets,  [Bao ACL Finding 2023] deletes part of the data(around 1.5%) in the ACOS dataset, which makes their results can not be compared directly.
> - As [Bao IJCAI 2022] utilizes external notation resources from other areas such as the trained AMR model, we directly adopt the performance of OTG without pretraining from [Bao IJCAI 2022] for a fair comparison, we will add the explanation into the paper once been accepted.
>
> > 3. It could be convincing to discuss case studies and error studies to highlight the effectiveness of each proposed component. For example, this paper mentions that the Element-level Graph Pre-training abandons the strategy of capturing the complex structure but focuses directly on the core elements. However, without case study, it is less convincing to figure it out. An example of case study can be found in “Graph pre-training for AMR parsing and generation”.
>
> #### **Response:** Due to page limitation, we finally decided not to put the case study on the paper. We believe your review's suggestion can surely help show the effectiveness of our method and we would like to add it to the appendix if accepted.
>
>
> > 3. In the main results, LLaMa-7B prominently performs better than ChatGPT. Generally, ChatGPT prominently outperforms LLaMa-7B, and 7B size is also not large enough. The main results show that after fine-tuning, LLaMa-7B can prominently performs better than ChatGPT. Could you introduce more details about the implementations and discuss more? e.g., the fine-tuning settings, the prompts used to these LLMs, should we use fine-tuned smaller LLMs rather than larger LLMs without fine-tuning?, etc.
>
> #### **Response:** We fine-tuned LLaMA -7B with Lora, and inferred opinion tree on ChatGPT with zero-shot learning. Based on our experiment result, we believe that in the NLP tasks whose targets are not close to natural language (e.g. opinion tree generation, AMR parsing), finetuning a smaller LLM might be better than direct inferring on the larges as they are not pre-trained to do so.

---

### Meta-Review · Area_Chair_4YeN · 2023-09-18

**Recommendation:** 4

**Metareview:**

The paper proposes two pre-training approaches to improve opinion tree generation for aspect-based sentiment analysis. The proposed method is sound validated by experiments. However, the experiments have been performed only on one dataset (ACOS) while more extensive evaluation is needed. Morever, the reviews highlighted that lack of extensive explanations on why the performance can be improved with the proposed method.

---

### Decision · Program_Chairs · 2023-10-07

**Decision:**

Accept-Findings

**Comment:**

The paper proposes two pre-training approaches to improve opinion tree generation for aspect-based sentiment analysis. The proposed method is sound validated by experiments. However, the experiments have been performed only on one dataset (ACOS) while more extensive evaluation is needed. Morever, the reviews highlighted that lack of extensive explanations on why the performance can be improved with the proposed method.